# Emetine Synergizes with Cisplatin to Enhance Anti-Cancer Efficacy against Lung Cancer Cells

**DOI:** 10.3390/ijms20235914

**Published:** 2019-11-25

**Authors:** Ti-Hui Wu, Shan-Yueh Chang, Yu-Lueng Shih, Tsai-Wang Huang, Hung Chang, Ya-Wen Lin

**Affiliations:** 1Graduate Institute of Medical Sciences, National Defense Medical Center, Taipei 11490, Taiwan; chestsurgerytsgh@gmail.com (T.-H.W.); leehornwok@gmail.com (S.-Y.C.); albreb@ms28.hinet.net (Y.-L.S.); 2Division of Thoracic Surgery, Department of Surgery, Tri-Service General Hospital, National Defense Medical Center, Taipei 11490, Taiwan; chi-wang@yahoo.com.tw (T.-W.H.); hung@ndmctsgh.edu.tw (H.C.); 3Division of Pulmonary and Critical Care, Department of Internal Medicine, Tri-Service General Hospital, National Defense Medical Center, Taipei 11490, Taiwan; 4Division of Gastroenterology, Department of Internal Medicine, Tri-Service General Hospital, National Defense Medical Center, Taipei 11490, Taiwan; 5Department and Graduate Institute of Microbiology and Immunology, National Defense Medical Center, Taipei 11490, Taiwan

**Keywords:** emetine, cisplatin resistance, lung cancer, Wnt/β-catenin

## Abstract

Cisplatin is still the primary therapeutic choice for advanced lung cancers without driver mutations. The occurrence of cisplatin resistance is a major clinical problem in lung cancer treatment. The natural extracted agent emetine reportedly has anticancer effects. This study aimed to explore the possible role of emetine in cisplatin resistance. We used cell viability, Western blot, and Wnt reporter assays to show that emetine suppresses proliferation, β-catenin expression, and Wnt/β-catenin signaling in non-small cell lung cancer (NSCLC). The synergism of emetine and cisplatin was assessed by constructing isobolograms and calculating combination index (CI) values using the Chou-Talalay method. Emetine effectively synergized with cisplatin to suppress the proliferation of cancer cells. Furthermore, nuclear β-catenin and cancer stem cell-related markers were upregulated in the cisplatin-resistant subpopulation of CL1-0 cells. Emetine enhanced the anticancer efficacy of cisplatin and synergized with cisplatin in the cisplatin-resistant subpopulation of CL1-0 cells. Taken together, these data suggest that emetine could suppress the growth of NSCLC cells through the Wnt/β-catenin pathway and contribute to a synergistic effect in combination with cisplatin.

## 1. Introduction

Lung cancer is the most common cause of cancer-related death worldwide [1]. Non-small cell lung cancer (NSCLC) accounts for ~75%–80% of lung cancer cases [2]. Only selected patients who have driver gene mutations are treated with targeted therapies. Platinum-based doublet chemotherapeutic regimens are still the primary therapeutic method for advanced NSCLC [3]. However, the five-year survival rate is still very low, and a therapeutic plateau has been reached, with most responders relapsing within six months [4,5]. Chemoresistance due to selection is the most common reason for the failure of cisplatin treatment [6].

Aberrant activation of the Wnt/β-catenin pathway plays a critical role in tumor initiation, progression, metastasis, and drug resistance in lung cancer [7,8,9,10,11]. Although mutations in β-catenin or APC are relatively rare in lung cancer, Wnt/β-catenin pathway activation is observed in approximately 50% of human NSCLC cell lines and primary tumors [12]. The Wnt/β-catenin pathway is also critical for the maintenance of cancer stem cells (CSCs) [11,13]. In a previous study, we used a Wnt-responsive Super-TOPflash (STF) luciferase reporter system in human hepatocellular carcinoma (HCC) cancer cell lines to screen compounds targeting Wnt signaling from the Library of Pharmacologically Active Compounds (LOPAC) [14]. Emetine dihydrochloride hydrate (emetine) was identified as one of the small molecule inhibitors of the Wnt/β-catenin pathway (Appendix A). Emetine, an emetic with a long history of use for the treatment of amoebiasis, is one of the natural active agents extracted from the plant *Psychotria ipecacuanha* [15]. In the 1970s, emetine was used in several phase I and II clinical trials by the National Cancer Institute to evaluate antitumor activity. However, it was not further pursued due to its significant toxicity during chronic usage [16]. Recently, emetine has been reported to exert antitumor effects in leukemia, ovarian carcinoma, bladder cancer, and human NSCLC via various pathways [17,18,19,20,21]. The reported mechanisms of emetine in treating cancers include inducing apoptosis in leukemia cell lines, downregulating Bcl-XL in ovarian carcinoma cells, inducing apoptosis and autophagy in bladder cancer cells, and regulating the ERK and p38 pathways in human NSCLC [17,18,19,20,21].

The purpose of this study was to evaluate the effect of emetine on human NSCLC cells and the cisplatin-resistant subpopulation of these cells. In addition, we sought to evaluate whether emetine could suppress the growth of NSCLC cells through the Wnt/β-catenin pathway and contribute to a synergistic effect in combination with cisplatin.

## 2. Results

### 2.1. Emetine Inhibits the Wnt/β-catenin Pathway, c-myc and Cyclin D1 in Human NSCLC Cells

First, we measured the endogenous β-catenin level in human NSCLC cells by Western blotting. The data showed that detectable expression of β-catenin was present in most of the NSCLC cells (Figure 1A). To determine whether emetine could inhibit the Wnt/β-catenin pathway, we analyzed the expression of β-catenin and its downstream targets, c-myc and cyclin D1, after NSCLC cells were treated with or without emetine. As the results indicated, β-catenin, c-myc and cyclin D1 were downregulated in NSCLC cells (CL1-0, CL1-5, A549, H1437, and H1355) after treatment with 120 nM emetine for 48 hours (Figure 1B). To further examine the role of emetine in the regulation of Wnt signaling, human NSCLC cells were treated with different doses of emetine for six hours, and the effect of emetine on Wnt signaling was evaluated by Super-TOPflash (STF) luciferase reporter assays. Emetine significantly decreased the transcriptional activity of TOPflash (M50)/FOPflash (M51) in CL1-0 and H1437 cells in a dose-dependent manner (Figure 1C).

### 2.2. Emetine Synergizes with Cisplatin to Enhance its Anticancer Efficacy in Human NSCLC Cells

To investigate whether emetine suppressed NSCLC cell proliferation effectively, cells were treated with increasing doses of emetine for 48 hours. The viability of these cells was evaluated by a 3-(4,5-dimethylthiazol-2-yl)-5-(3-carboxymethoxyphenyl)-2-(4-sulfophenyl)-2H-tetrazolium (MTS) assay and was found to be reduced in an emetine dose-dependent manner (Appendix A). The 50 percent inhibitory concentration (IC_50_) values of emetine in human NSCLC cells were in the nanomolar range (Appendix A). Cisplatin has been used as the first-line drug for chemotherapeutic administration in cases of advanced and metastatic lung cancer for decades. Thus, we next examined whether emetine could cooperate with cisplatin to enhance its antiproliferative effects. We treated human NSCLC cells with a constant-ratio combination of emetine and cisplatin for 48 hours and assessed the synergism by constructing isobolograms and calculating combination index (CI) values using the Chou-Talalay method [22,23]. The results demonstrated that the combination of emetine and cisplatin was synergistic in A549, CL1-0, CL1-5, H1355, H1437, H358, and H647 cells but was additive in H1299 cells (Figure 2). Specific CI values at the ED50, ED75 and ED90 are shown in Appendix A. Collectively, these results suggest that emetine suppresses human NSCLC cell proliferation synergistically in combination with cisplatin.

### 2.3. Low-Dose Emetine Sensitizes Lung Cancer Cells to Cisplatin

Emetine causes concern regarding its significant toxicity, particularly the muscle weakness and cardiotoxicity associated with its chronic use [16,24,25,26,27,28,29]. To prevent these possible adverse effects, which might impede its clinical use, we tested whether low-dose emetine also performed well in combination with cisplatin. Human NSCLC cells were treated with increasing concentrations of cisplatin along with a fixed dose of 40 nM emetine for 48 hours. The addition of 40 nM emetine resulted in an approximately 2–7-fold decrease in the IC_50_ of cisplatin required to suppress NSCLC cell proliferation (Figure 3, Appendix A). That is, even at this concentration that was much lower than the maximum safe therapeutic concentration and thus abolished the concern of possible adverse effects, emetine still showed good efficacy in inhibiting tumor cell proliferation.

### 2.4. Nuclear β-catenin and Stemness-Related Markers are Upregulated in the Cisplatin-resistant Subpopulation of CL1-0 cells

As previously reported, the resistance of cancer cells to cisplatin due to natural selection during treatment is the most common reason for the failure of cisplatin treatment [6]. To further recapitulate the characteristics of cisplatin-resistant conditions, we generated cisplatin-resistant cells from parental CL1-0 NSCLC cells via the pulse method [30]. The generated CL1-0 cisplatin-resistant subpopulation of cells (CL1-0/CDDP) showed a 2.17-fold higher IC_50_ value for cisplatin than the parental counterparts (Figure 4A). CL1-0/CDDP cells grew more slowly than the parental cells (Figure 4B). Moreover, cisplatin-resistant human NSCLC cells have been reported to display a stem-like signature [30], and cancer stem cells (CSCs) have been suggested to be responsible for drug resistance [11,13]. We then evaluated the stemness characteristics of CL1-0/CDDP cells and the parental CL1-0 cells by a sphere formation assay. Cells were grown in an ultralow attachment, serum-starved culture system for 30 days to generate spherical colonies. The CL1-0/CDDP cells exhibited larger spherical colony diameters than the parental CL1-0 cells (Figure 4C). To further assess Wnt//β-catenin signaling in the cisplatin-resistant subpopulation, the β-catenin level was evaluated by Western blotting and immunocytochemical staining. Stemness-related markers and epithelial-mesenchymal transition (EMT) markers were evaluated by quantitative real-time RT-PCR. The results demonstrated that both the total and nuclear expression of β-catenin were elevated in CL1-0/CDDP cells (Figure 4D) and that the elevated β-catenin expression was distributed in the cytoplasm and nucleus (Appendix A). The mRNA expression levels of *NANOG*, *OCT4*, and *CDH2* were increased in CL1-0/CDDP cells. However, there was no difference in the mRNA expression levels of *MYC*, *CCND1*, and *EPCAM* (Figure 4E). Taken together, these results demonstrated that upregulation of stemness-related and EMT-related genes and an accumulation of nuclear β-catenin occurred in the CL1-0 cisplatin-resistant subpopulation cells.

### 2.5. Emetine Enhances The Anticancer Efficacy of Cisplatin and Synergizes with Cisplatin in the Cisplatin-Resistant Subpopulation of CL1-0 Cells

To investigate whether the synergistic antiproliferative effect of the combination of emetine and cisplatin occurred in cisplatin-resistant cells, we treated CL1-0/CDDP cells with a constant-ratio combination of emetine and cisplatin or with increasing concentrations of cisplatin along with a fixed low dose of 40 nM emetine for 48 hours. The results showed that the combination of emetine and cisplatin was synergistic (Figure 5A), with a mean CI of 0.627 (Appendix A) and that the addition of low-dose emetine resulted in an approximately 2.33-fold decrease in the IC_50_ of cisplatin required to suppress the proliferation (Figure 5B) of CL1-0/CDDP cells. Taken together, these results suggested that emetine is still effective even in cisplatin-resistant NSCLC cells.

### 2.6. Emetine Inhibits the Wnt/β-catenin Pathway in the Cisplatin-Resistant Subpopulation of CL1-0 Cells

Our data demonstrated that emetine inhibited the Wnt/β-catenin pathway in human NSCLC cells. Then, we examined whether emetine could suppress the Wnt/β-catenin pathway in cisplatin-resistant cells. The data showed downregulation of β-catenin, c-myc and cyclin D1 after CL1-0/CDDP cells were treated with 120 nM emetine for 48 hours (Figure 6A). Furthermore, emetine significantly decreased the transcriptional activity of TOPflash/FOPflash in CL1-0/CDDP cells in a dose-dependent manner in STF luciferase reporter assays (Figure 6B). Previous studies have reported that emetine suppresses the expression of HIF-1α, which plays a central role in tumor progression, invasion, and metastasis [31,32]. To clarify the possible role of HIF-1α in the antitumor effect of emetine in human NSCLC cells, CL1-0/CDDP cells were treated with 10 μM cisplatin only, 40 nM emetine only, or a combination of cisplatin and emetine. These data showed that the expression of HIF-1α was not affected after cancer cells were treated with these drugs (Appendix A) and suggested that HIF-1α might not be involved in the synergistic mechanism of emetine.

## 3. Discussion

Platinum-based systemic chemotherapy is still the mainstay of treatment in lung cancer despite advances in targeted therapies and immunotherapies. However, drug resistance remains a major obstacle to treatment in lung cancer patients, leading to tumor recurrence and disease progression [6]. Additionally, cisplatin is associated with a number of serious toxicities and side effects [33]. Combining novel agents targeting specific resistance pathways with standard chemotherapy could reveal some promising strategies to overcome chemotherapeutic resistance in lung cancer [34] and to decrease the cisplatin therapeutic dose to reduce the possible side effects. In this study, we demonstrated that the repurposed drug emetine synergized with cisplatin to enhance its anticancer effect under cisplatin-naïve and cisplatin-resistant conditions. The anticancer effect of emetine was mediated through the inhibition of the Wnt/β-catenin pathway, and emetine was efficient even at low doses. However, the development of a brand-new drug is unfavorable in terms of time frame, costs and effort. Drug repurposing is one strategy advanced to shorten this time frame, decrease costs, and improve success rates. Many drugs approved for a different disease already have well-established safety profiles, optimal dosing regimens, and acceptable toxicity. The combination of repurposed drugs with current therapeutics has the advantage of providing an economical, safe and efficacious approach to overcome drug resistance, reduce side effects and prolong survival in cancer patients [35,36,37]. Emetine is one of the repurposed drugs used to treat cancers long ago, but its use was limited by its dose-dependent toxicity in chronic use [16]. In this study, we demonstrated that emetine’s antitumor effect occurred at nanomolar concentrations in human NSCLC cells, and the addition of a low dose (40 nM) of emetine further sensitized human NSCLC cells to cisplatin (Figure 2 and Figure 3). At this therapeutic concentration, we could decrease the toxicity of emetine, which is an important consideration for its practical clinical use. Additionally, we demonstrated that emetine synergized with cisplatin in seven of eight human NSCLC cell lines, with a mean CI ranging from 0.492 to 1.013 (Appendix A). With this synergism, combination treatment with emetine and cisplatin could enhance cisplatin’s antitumor effect, reduce cisplatin’s toxicity and delay the acquisition of cisplatin resistance that might be encountered. In this report, we demonstrated that emetine suppresses tumor cell proliferation by inhibiting Wnt//β-catenin signaling and its target genes, such as c-myc and cyclin D1 (Figure 1). To our knowledge, no report has demonstrated that the mechanism of action of emetine is mediated through the inhibition of the Wnt//β-catenin pathway.

The Wnt/β-catenin pathway plays a critical role in cancer initiation, development, and metastasis in various human cancer types, including lung cancer [7,8,9,10,11]. In lung cancer, dysregulation of Wnt signaling has been found [38], and overexpression of Wnt proteins (Wnt1 and Wnt5a) was notably linked to unfavorable overall survival in lung cancer patients [39]. In addition, constitutive Wnt pathway activation is associated with tumor recurrence and poor prognosis in NSCLC patients [9,40,41,42,43,44,45]. The Wnt/β-catenin pathway has also been reported to play a key role in cisplatin resistance in various human cancer types, including lung cancer [46,47,48,49,50,51,52,53,54,55,56,57,58,59,60,61]. Cancer stem cells (CSCs) are associated with chemoresistance, and Wnt signaling is one of their regulators [11,13]. Therefore, targeting Wnt signaling could not only enhance cisplatin’s antitumor efficacy, most importantly, but also suppress the occurrence of CSCs. Wnt/β-catenin signaling plays an important role not only in driver mutation-naïve patients but also in patients with driver mutations. Blakely and his colleagues performed a comparative genomic analysis of cell-free DNA from large cohorts of patients with stage III/V EGFR-mutated NSCLC. They showed that genetic alterations in Wnt/β-catenin pathways were a common co-occurrence in the EGFR-mutated patient population and that coalterations in Wnt/β-catenin pathways were associated with a poor response to EGFR-TKIs [62]. Therefore, adding emetine to TKI therapy might improve the overall response rate and decrease the occurrence of TKI resistance. However, proving this hypothesis will require further study.

Our data also demonstrated that the cisplatin-resistant subpopulation of CL1-0 cells exhibited upregulated β-catenin, Nanog, Oct-4 and CDH2 signaling (Figure 4). This finding was the same as that in Teng and Wang’s report, namely that Nanog and Oct-4 levels were significantly elevated in drug-resistant lung cancer cells [63,64] and in Wang and Zhang’s report, which showed that CDH2 contributes to acquired drug resistance in NSCLC [65,66]. Considering these findings together with the increased tumor sphere formation ability of the cisplatin-resistant subpopulation of CL1-0 cells (Figure 4C), we verified that chemoresistant NSCLC cells were associated with the acquisition of CSC-like properties. However, there is no difference in the expression of *MYC, CCND1* and *EPCAM* in the cisplatin-resistant subpopulation of CL1-0 cells. This discrepancy might be due to the heterogeneity of chemoresistant NSCLC cells [63,64,65,66]. Based on our data, we found that cisplatin-resistant subpopulation of CL1-0 cells grew much more slowly than original cells. From the cell morphology, we observed that cisplatin-resistant subpopulation of CL1-0 cells aggregated much more than control. Therefore, we suggested that cisplatin-resistant subpopulation of CL1-0 cells might be dormant and more quiescent than in the original cells. Last and most importantly, in this study, we revealed that emetine was effective under cisplatin-resistant conditions and synergized with cisplatin, with a mean CI of 0.627 in the cisplatin-resistant subpopulation of CL1-0 cells (Figure 5 and Appendix A). Thus, emetine might serve as a salvage treatment in cisplatin-resistant patients to restore the antitumor response.

Our study has some limitations. First, we evaluated the function of emetine only in human NSCLC cells. No further studies were performed in an animal model, for two major reasons. The first reason is that synergism is easy to define in a cell model but difficult to define in an animal model using the Chou-Talalay method. The second reason is that in animal models, cells grow in a hypoxic environment, which could activate the HIF pathway, which might in turn interfere with the effect of emetine. For these two reasons, we studied the function of emetine only in human NSCLC cells. The second limitation is that the cisplatin-resistant subpopulation of CL1-0 cells was obtained by drug selection. These subpopulations were heterogeneous clones, and non-Wnt/β-catenin pathways likely also contributed to cisplatin resistance. In the future, we aim to explore how emetine regulates the level of β-catenin and, furthermore, decipher additional molecular mechanisms regarding how emetine increases the sensitivity of cisplatin-resistant cells to cisplatin using a more comprehensive strategy. Our finding that emetine’s antitumor effect was mediated through the inhibition of the Wnt/β-catenin pathway is only one of the possible mechanisms for overcoming cisplatin resistance. In summary, emetine suppressed lung cancer cell proliferation and synergized with cisplatin to overcome drug resistance. Its effect was partially mediated through the Wnt/β-catenin pathway.

## 4. Materials and Methods 

### 4.1. Cell Lines and Drug Treatment

Eight human lung cancer cell lines (A549, CL1-0, CL1-5, H1299, H1355, H1437, H358, and H647) were used in this study. They were a kind gift from Professor Yi-Ching Wang (National Cheng Kung University, Taiwan, China). Cells were cultured in Roswell Park Memorial Institute (*RPMI*) 1640 medium (GIBCO, Gaithersburg, MD, USA) supplemented with 2 mM L-glutamine, 1% nonessential amino acids (NEAA), 10% heat-inactivated fetal bovine serum (FBS), 100 U/mL penicillin and 100 μg/mL streptomycin (GE Healthcare Life Sciences, Chicago, IL, USA). All cell cultures were grown as monolayer cultures and maintained in a humidified atmosphere containing 5% CO_2_ in air at 37 °C.

### 4.2. Drugs

Cisplatin (*cis*-diammineplatinum(II) dichloride) and emetine (emetine dihydrochloride hydrate) were obtained from Sigma-Aldrich (St Louis, MO, USA). Cisplatin and emetine were dissolved in distilled water. Aliquots were stored at −20 °C for up to a maximum of 3 months and thawed immediately before use.

### 4.3. Establishment of the Cisplatin-Resistant Cell Subpopulations

Cisplatin-resistance subpopulations of CL1-0 cells (CL1-0/CDDP) were derived from CL1-0 parental cells by the pulse method [30] as follows: CL1-0 cells were treated with cisplatin (Sigma-Aldrich) at the IC_50_ concentration for 48 h. The medium was removed, and cells were allowed to recover to approximately 70% confluence. This cycle was repeated, and the IC_50_ concentrations were reassessed in the resistant cells every month. Cells were then treated with cisplatin at the new IC_50_ concentration for another month. This development process was repeated for approximately 6 months. Cells were then maintained continuously in the absence of cisplatin.

### 4.4. Cell Viability Assay

Cells (5 × 10^3^) were seeded in 96-well plates and allowed to adhere overnight at 37 °C. These cells were treated with distilled water or emetine (Sigma-Aldrich) at the indicated concentrations for 48 hours. Cell viability was measured by an MTS assay (Sigma-Aldrich). All experiments were performed in triplicate.

### 4.5. Isobologram and the Combination Index (CI)

Cells were plated in triplicate wells at 5 × 10^3^ cells per well in growth medium in 96-well plates and incubated overnight. Cells were treated with the indicated concentrations of emetine alone, cisplatin alone, a constant-ratio combination of emetine and cisplatin, or the indicated concentrations of cisplatin with 40 nM emetine for 48 hours. Cell viability was measured using an MTS assay. The synergism of emetine and cisplatin was assessed by constructing isobolograms and calculating combination index (CI) values using the Chou-Talalay method [22,23] in CompuSyn software version 3.0.1 (ComboSyn, Inc., Paramus, NJ, USA). Synergistic effects are indicated by values below the line in the isobologram or as a CI of <1; additive effects are indicated by values close to the line in the isobologram or as a CI of 1, and antagonistic effects are indicated by values above the line in the isobologram or as a CI of >1.

### 4.6. Western Blot Analysis

Protein extraction was performed based on our previous report [14]. Primary antibodies were diluted 1:2000 and incubated overnight at 4 °C. Secondary antibodies diluted 1:5000 were added and incubated at room temperature for 1 h. Signals were detected using ECL detection reagent (Millipore Corporation, Burlington, MA, USA) following the manufacturer’s instructions. The primary antibodies used were as follows: mouse anti-β-catenin, mouse anti-lamin A/C (BD Biosciences, San Jose, CA, USA), mouse anti-c-myc, mouse anti-cyclin D1, rabbit anti-HDAC1, rabbit anti-HIF-1α and rabbit anti-β-actin (GeneTex, Irvine, CA, USA). Horseradish peroxidase-conjugated rabbit anti-mouse or goat anti-rabbit secondary antibodies (GeneTex) were used as appropriate. Detailed information is given in the Appendix A.

### 4.7. TCF/LEF Luciferase Assay

The pGL4.21-TOPflash and pGL4.21-FOPflash vectors were generated and described in our previous report [14]. CL1-0, H1437, and CL1-0/CDDP cells were transfected with the pGL4.21-TOPflash vector and incubated for 2 days. Then, cells were treated with the indicated concentration of emetine for 6 hours. Firefly and Renilla luciferase activities were measured with a Dual-Glo Luciferase Assay System (Promega, Madison, WI, USA). Detailed information is given in the Appendix A. 

### 4.8. Spheroid Formation Ability

Cells were collected and washed to remove serum and suspended in serum-free RPMI 1640 medium supplemented with 100 IU/mL penicillin, 100 μg/ml streptomycin, 20 ng/mL human recombinant epidermal growth factor (hrEGF), 10 ng/mL human recombinant basic fibroblast growth factor (hrbFGF), 2% B27 supplement without vitamin A, and 1% N2 supplement (Invitrogen, Carlsbad, CA, USA). Cells were subsequently cultured in ultralow-attachment 96-well plates (Corning Inc., Corning, NY, USA) at a density of 200 cells/well. The spheres were then collected and counted after 30 days.

### 4.9. Immunocytochemical (ICC) Staining

CL1-0 and CL1-0/CDDP cells were seeded on 12-mm glass coverslips and cultured overnight. Cells were washed with phosphate-buffered saline (PBS) and fixed in 4% formaldehyde solution (Sigma-Aldrich). The primary antibody against mouse anti-β-catenin (BD Biosciences) was added and incubated overnight at 4 °C. Cells were washed and stained with Hoechst 33342 (bisbenzimide H33342 trihydrochloride) (Sigma-Aldrich) to visualize nuclei. Stained cells on coverslips were mounted to slides using Fluoromount Aqueous Mounting Medium (Sigma-Aldrich). Fluorescence images were acquired with an LSM 880 confocal laser scanning microscope with Airyscan (Carl Zeiss Inc., Oberkochen, Germany). FITC-conjugated anti-mouse IgG (Bethyl Laboratories, Montgomery, TX, USA) and DyLight 594-labeled anti-rabbit IgG (GeneTex) secondary antibodies were used as appropriate. Detailed information is given in the Appendix A.

### 4.10. Nuclear Extraction

Nuclear extracts were prepared using a Nuclear Extraction Kit 2900 (Millipore Corporation) according to the manufacturer’s instructions. Detailed information is given in the Appendix A.

### 4.11. RNA Extraction and Real-Time Quantitative Polymerase Chain Reaction

Total RNA was isolated from CL1-0 and CL1-0/CDDP cells using TRIzol reagent (Invitrogen) according to the manufacturer’s instructions. RNA was reverse transcribed to cDNA using Superscript II reverse transcriptase (Invitrogen). The RT-PCR primer sequences used for this study are shown in Appendix A. The relative gene expression was quantified via real-time RT-PCR using SYBR Green I reagents. *PBGD* was used as the internal control. All reactions were performed in triplicate.

### 4.12. Statistical Analysis

Statistical analyses were performed using GraphPad Prism software (Version 4.03; GraphPad Software Inc., La Jolla, CA) or SPSS software (IBM SPSS Statistics 21; Asia Analytics Taiwan Ltd., Taipei, Taiwan). The data are expressed as the means ± SEMs. *p* < 0.05 was considered significant.

## 5. Conclusions

In conclusion, we demonstrated that emetine synergizes with cisplatin to enhance its anticancer effect and overcomes cisplatin resistance partially by suppressing the Wnt/β-catenin pathway in human NSCLC cells. This emetine and cisplatin combination therapy may offer a potential therapeutic strategy for NSCLC.

## Figures and Tables

**Figure 1 ijms-20-05914-f001:**
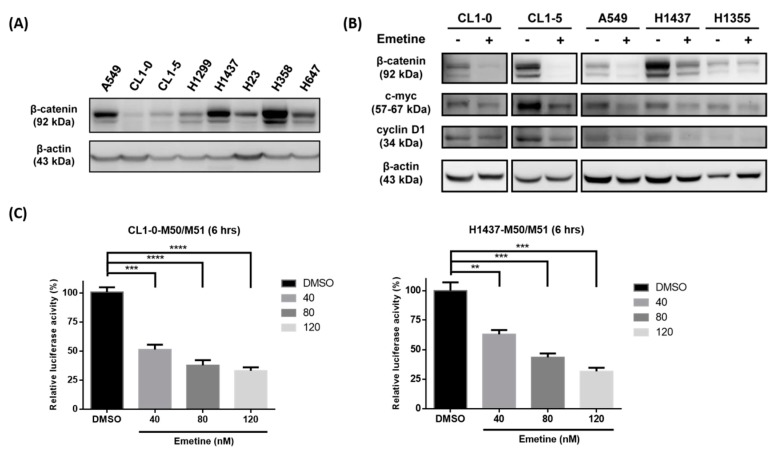
Emetine inhibits the Wnt/β-catenin pathway, c-myc and cyclin D1 in human non-small cell lung cancer (NSCLC) cells. (**A**) The endogenous expression of total β-catenin in A549, CL1-0, CL1-5, H1299, H23, H358, and H647 human NSCLC cells was examined by Western blotting. β-Actin was used as the internal control. (**B**) CL1-0, CL1-5, A549, H1437, and H1355 human NSCLC cells were treated with or without 120 nM emetine for 48 hours. The protein expression of β-catenin, c-myc, and cyclin D1 was examined by Western blotting. β-Actin was used as the internal control. (**C**) The TOPflash (M50) reporter containing wild-type TCF/LEF binding sites produced a high level of transcriptional activity. The FOPflash (M51) reporter containing mutated TCF/LEF binding sites was used as the negative control. The relative luciferase activity of TOPflash/FOPflash was analyzed after 6 h of treatment with DMSO or the indicated concentration of emetine in the CL1-0 and H1437 cell lines. The data are expressed as the means ± SDs from three independent experiments. ** *p* < 0.01, *** *p* < 0.001, **** *p* < 0.0001 (Student’s *t*-test).

**Figure 2 ijms-20-05914-f002:**
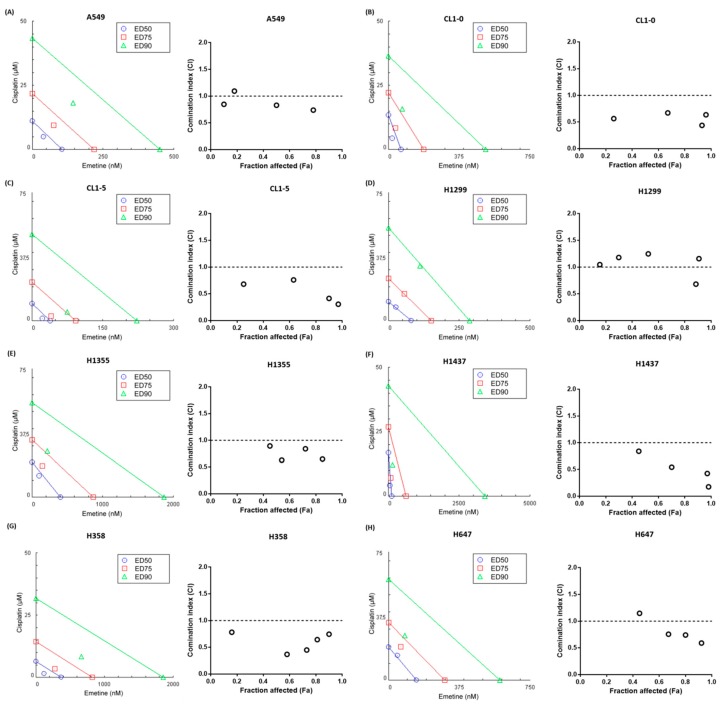
Emetine synergizes with cisplatin to enhance its anticancer efficacy in human NSCLC cells. Cell lines were exposed to a constant-ratio combination of emetine and cisplatin for 48 hours. Cell viability was measured using an MTS assay. The synergism of emetine and cisplatin was assessed by constructing isobolograms and calculating combination index (CI) values using the Chou-Talalay method in CompuSyn software (ComboSyn, Inc.). (**A**) A549, (**B**) CL1-0, (**C**) CL1-5, (**D**) H1299, (**E**) H1355, (**F**) H1437, (**G**) H358, (**H**) H647. For each cell line, the isobologram (left) shows the drug combination at the ED_50_, ED_75_ and ED_90_ effect levels. The Fa-CI plot (right) shows the combination index as a function of the fraction affected. Synergistic effects are indicated by values below the line in the isobologram or as a CI of < 1 in the Fa-CI plot; additive effects are indicated by values close to the line in the isobologram or as a CI of 1, and antagonistic effects are indicated by values above the line in the isobologram or as a CI of >1. The data are representative of three independent experiments performed in triplicate. ED_50_, 50% effective dose. ED_75_, 75% effective dose. ED_90_, 90% effective dose.

**Figure 3 ijms-20-05914-f003:**
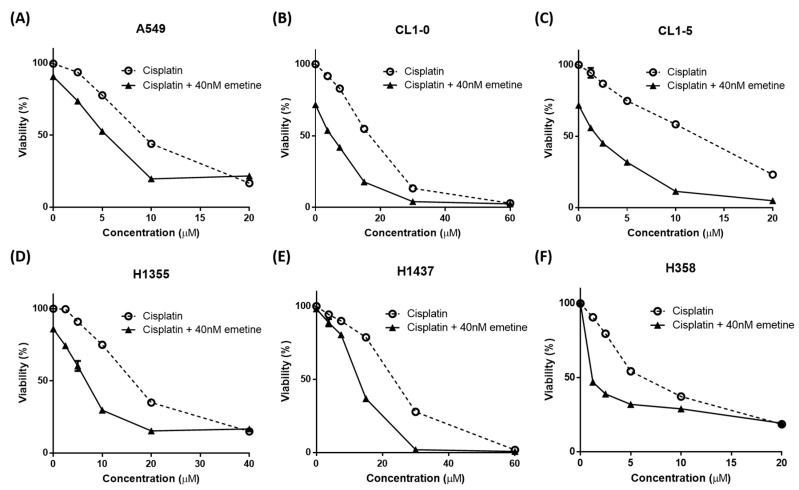
Low-dose emetine sensitizes lung cancer cells to cisplatin. Human NSCLC cells were treated with increasing concentrations of cisplatin with or without a fixed dose of 40 nM emetine for 48 h. Cell viability was measured using an MTS assay. (**A**) A549, (**B**) CL1-0, (**C**) CL1-5, (**D**) H1355, (**E**) H1437, (**F**) H358. The data are shown as the means ± SEMs from three independent experiments performed in triplicate.

**Figure 4 ijms-20-05914-f004:**
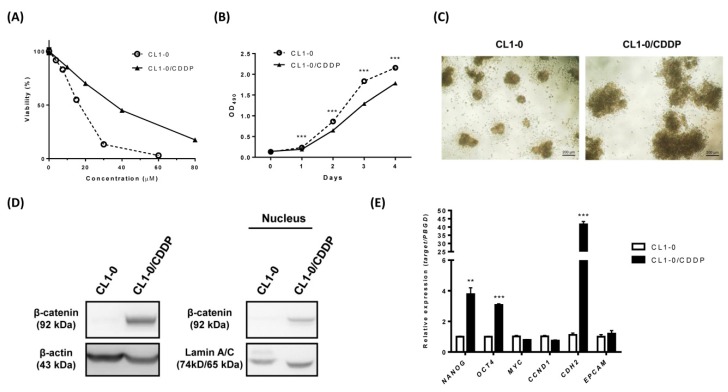
Nuclear β-catenin and cancer stem cell-related markers are upregulated in the cisplatin-resistant subpopulation of CL1-0 cells. (**A**) CL1-0 cells and the cisplatin-resistant subpopulation of CL1-0 (CL1-0/CDDP) cells were treated with increasing concentrations of cisplatin for 48 h. Cell viability was measured using an MTS assay. The IC_50_ values of cisplatin in CL1-0/CDDP and CL1-0 cells were 33.35 µM and 15.34 µM, respectively. (**B**) Cell viability was analyzed by an MTS assay in CL1-0 cells and the cisplatin-resistant subpopulation of CL1-0 cells. (**C**) Phase contrast images of CL1-0 cells and the cisplatin-resistant subpopulation of CL1-0 (CL1-0/CDDP) cells cultured in a 96-well ultralow-attachment plate under anchorage-independent, serum-free conditions with supplementation of growth factor and B27 for 30 days. (**D**) Total and nuclear expression levels of β-catenin in CL1-0 cells and the cisplatin-resistant subpopulation of CL1-0 (CL1-0/CDDP) cells were measured by Western blotting. β-Actin and Lamin A/C were used as internal controls. (**E**) The mRNA expression levels of *NANOG, OCT4, MYC, CCND1, CDH2,* and *EPCAM* in CL1-0 cells and the cisplatin-resistant subpopulation of CL1-0 cells were measured by qRT-PCR and were normalized using the *PBGD* gene. The data are expressed as the means ± SDs from three independent experiments. ** *p* < 0.01, *** *p* < 0.001 (Student’s *t*-test).

**Figure 5 ijms-20-05914-f005:**
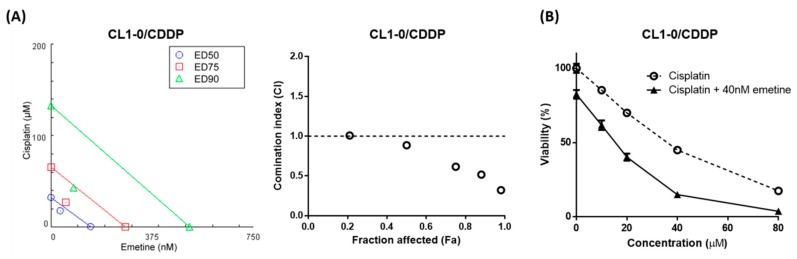
Emetine enhances the anticancer efficacy of cisplatin and synergizes with cisplatin in the cisplatin-resistant subpopulation of CL1-0 cells. (**A**) Cells in the cisplatin-resistant subpopulation of CL1-0 cells (CL1-0/CDDP) were exposed to a constant-ratio combination of emetine and cisplatin for 48 hours. Cell viability was measured using an MTS assay. The synergism of emetine and cisplatin was assessed by constructing an isobologram (left) and a Fa-CI plot (right) using the Chou-Talalay equation in CompuSyn software (ComboSyn, Inc.). (**B**) Cells in the cisplatin-resistant subpopulation of CL1-0 cells (CL1-0/CDDP) were treated with increasing concentrations of cisplatin without or with the combination of 40 nM emetine for 48 hours. Cell viability was measured using an MTS assay. The IC_50_ values of cisplatin without and with 40 nM emetine were 33.35 μM and 14.27 μM, respectively. The data are shown as the means ± SEMs of 3 independent experiments performed in triplicate.

**Figure 6 ijms-20-05914-f006:**
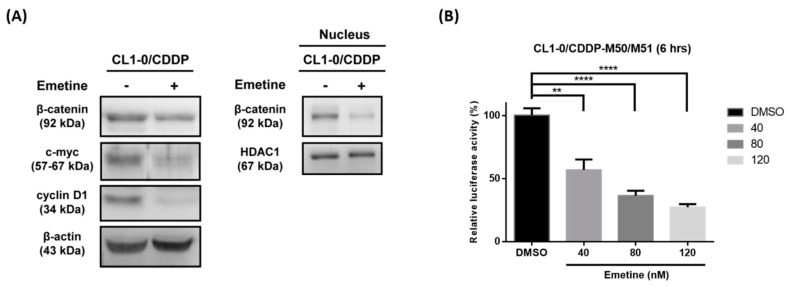
Emetine inhibits the Wnt/β-catenin pathway in the cisplatin-resistant subpopulation of CL1-0 cells. (**A**) Cells in the cisplatin-resistant subpopulation of CL1-0 cells (CL1-0/CDDP) were treated with or without 120 nM emetine for 48 hours. The total protein expression of β-catenin, c-myc, and cyclin D1 and the nuclear expression of β-catenin were assessed by Western blotting. β-Actin and HDAC1 were used as internal controls. (**B**) The TOPflash (M50) reporter containing wild-type TCF/LEF binding sites produced a high level of transcriptional activity. The FOPflash (M51) reporter containing the mutated TCF/LEF binding sites was used as the negative control. The relative luciferase activity of TOPflash/FOPflash was analyzed after six hours of treatment with DMSO or the indicated concentration of emetine in the cisplatin-resistant subpopulation of CL1-0 cells (CL1-0/CDDP). The data are expressed as the means ± SDs from three independent experiments. ** *p* < 0.01, **** *p* < 0.0001 (Student’s *t*-test).

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
