# Peer review of "Emetine Synergizes with Cisplatin to Enhance Anti-Cancer Efficacy against Lung Cancer Cells"

_ijms, 2019, doi:10.3390/ijms20235914_

Round 1

Reviewer 1 Report

In the manuscript, the authors present an in vitro experimental study regarding the efficacy of emetine in enhancing the therapeutic effect of cisplatin in patients with non-small cell lung cancer. Emetine is an well recognized old drug and in the last years there are many manuscripts that studied the antitumor effect of emetine in various types of cancers (ovarian, bladder). The problem with this drug is its cardiotoxicity, that can limit its clinical use. In the study, the authors used 8 human lung cancers cells lines. It is an interesting study and the methodology, the results and the conclusions of the manuscript are pertinent. In my opinion this manuscript can be published in that form. I think that this manuscript is is one of the few studies published in the literature on this topic. Also, there are some grammar and spelling errors in English, that is why the English language must be revised.  

Author Response

Point-by-point responses to the reviewer’s comments on the manuscript

RE: ijms-631417 entitled “Emetine Synergizes with Cisplatin to Enhance Anti-cancer Efficacy against Lung Cancer Cells”

We appreciate for the comments and suggestions of the reviewers. We have inserted the revisions with highlight or tracking changes on the modified version of ijms-631417 according to the reviewers’ comments.

Reviewer 1

In the manuscript, the authors present an in vitro experimental study regarding the efficacy of emetine in enhancing the therapeutic effect of cisplatin in patients with non-small cell lung cancer. Emetine is an well recognized old drug and in the last years there are many manuscripts that studied the antitumor effect of emetine in various types of cancers (ovarian, bladder). The problem with this drug is its cardiotoxicity, that can limit its clinical use. In the study, the authors used 8 human lung cancers cells lines. It is an interesting study and the methodology, the results and the conclusions of the manuscript are pertinent. In my opinion this manuscript can be published in that form. I think that this manuscript is is one of the few studies published in the literature on this topic. Also, there are some grammar and spelling errors in English, that is why the English language must be revised.

Response:
We greatly appreciate the positive comments from you. We have used a professional English editing service by MDPI to check the revised manuscript. Attached please find the editing certificate. The revised manuscript has been approved by all authors.

Reviewer 2 Report

The introduction section needs to further elaborated.

Line 51-54 about emetine being an inhibitor of the pathway Wnt/beta-catenin pathway has unpublished data. If the authors are the one who are first to report it they should either provide the data as a part of the main manuscript or supplementary information. If not then they would need to provide evidence or reference for it.

It is essential to provide the complete blots in the supplementary files and the replicates of Western blot. 

Also most of the western blot images do not have the molecular weight marked and needs to be done.

Please provide complete figure 4 D for the western blot.

Why the authors have not validated or confirmed the result obtained in-vitro by performing animal studies.

Reviewer 3 Report

The role of emetine in overcoming cisplatin resistance has been previously explored in several cancer types including ovarian carcinoma cells, bladder cancer and leukemia cells. Thus, the idea of ementine as a chemosensitizer in cancer is not novel.  

The expression of β-catenin was below detection limit in most of the cell lines, and this certainly could not be classified as detectable expression as claimed by the authors (Fig. 1A). Western blots presented in Fig. 1B, especially for c-myc and cyclin D1 are extremely poor in quality and intensity (weak bends) and do not provide solid evidence for conclusions presented under 2.1. section.

The IC50 values listed in the legend of Fig. 3 should be separately presented in the table.

Section 2.4. Line 150: Immunohistochemical staining is not performed in cells but tissues; this result is not presented in the manuscript. It has been previously found that subpopulation of cisplatin-resistant non-small-lung cancer cells have stem cell-like features. Thus, the results presented in section 2.4. do not provide any novel information. In particular, lack of explanation for the results observed with MYC, CCND1 and EPCAM are considered a minus.

The authors clearly showed that low nanomolar concentration of ementine reverses resistance of non-small-lung cancer cells to cisplatin and that Wnt-β catenin signaling pathway mediates cisplatin resistance in these cells. The latter signaling pathway has been previously established as a regulator of cisplatin resistance in lung adenocarcinoma cells.

One of the weaknesses of this work is the lack of additional mechanistic studies to provide a more comprehensive and novel insight into the mechanisms by which ementine increases the sensitivity of cisplatin-resistant cells to cisplatin.  Molecular basis of observed chemosensitization effects is poorly presented. 

Round 2

Reviewer 2 Report

The authors have made effort to address my comments. I recommend the publication of this manuscript.

Reviewer 3 Report

In spite of revisions made by the authors, scientific relevance and novelty of this paper are still limited.